# Time to Recovery from Systolic Dysfunction Correlates with Left Ventricular Fibrosis in Arrhythmia-Induced Cardiomyopathy

**DOI:** 10.3390/life14030330

**Published:** 2024-03-01

**Authors:** Christian Schach, Daniel Lavall, Nicola Voßhage, Thomas Körtl, Christine Meindl, Ekrem Ücer, Okka Hamer, Lars S. Maier, Rolf Wachter, Samuel Sossalla

**Affiliations:** 1Klinik und Poliklinik für Kardiologie, Universitäres Herzzentrum Regensburg, Universitätsklinikum Regensburg, Franz-Josef-Strauss-Allee 11, 93042 Regensburg, Germanysamuel.sossalla@innere.med.uni-giessen.de (S.S.); 2Klinik und Poliklinik für Kardiologie, Universitätsklinikum Leipzig, Liebigstrasse 20, 04103 Leipzig, Germany; 3Medizinische Klinik I, Justus-Liebig-Universität Gießen, Klinikstr. 33, 35392 Gießen, Germany; 4Institut für Röntgendiagnostik, Universitäres Herzzentrum Regensburg, Universitätsklinikum Regensburg, Franz-Josef-Strauß-Allee 11, 93053 Regensburg, Germany; 5Abt. Kardiologie, Kerckhoff Klinik, Bad Nauheim & German Center for Cardiovascular Research (DZHK), Partner Site RheinMain, 61231 Bad Nauheim, Germany

**Keywords:** atrial fibrillation, tachymyopathy, heart failure, left ventricular dysfunction, rhythm control, cardiac MRI, late gadolinium enhancement

## Abstract

Background: Arrhythmia-induced cardiomyopathy (AIC) is characterized by the reversibility of left ventricular (LV) systolic dysfunction (LVSD) after rhythm restoration. This study is a cardiac magnetic resonance tomography substudy of our AIC trial with the purpose to investigate whether left ventricular fibrosis affects the time to recovery (TTR) in patients with AIC. Method: Patients with newly diagnosed and otherwise unexplainable LVSD and tachyarrhythmia were prospectively recruited. LV ejection fraction (LVEF) was measured by echocardiography at baseline and 2, 4, and 6 months after rhythm control, and stress markers were assessed. After initial rhythm control, LV fibrosis was assessed through late gadolinium enhancement (LGE). Patients were diagnosed with AIC if their LVEF improved by ≥15% (or ≥10% when LVEF reached ≥50%). Non-responders served as controls (non-AIC). Results: The LGE analysis included 39 patients, 31 of whom recovered (AIC). LV end-systolic diameters decreased and LVEF increased during follow-up. LV LGE content correlated positively with TTR (r = 0.63, *p* = 0.003), with less LGE favoring faster recovery, and negatively with ΔLVEF (i.e., LVEF at month 2 compared to baseline) as a marker of fast recovery (r = −0.55, *p* = 0.012), suggesting that LV fibrosis affects the speed of recovery. Conclusion: LV fibrosis correlated positively with the time to recovery in patients with AIC. This correlation may help in the estimation of the recovery period and in the optimization of diagnostic and therapeutic strategies for patients with AIC.

## 1. Introduction

Atrial fibrillation (AF) is the most prevalent heart rhythm disorder and can result in arrhythmia-induced cardiomyopathy (AIC). The coexistence of AF and left ventricular (LV) systolic dysfunction (LVSD) is frequent [1], which may indicate under-recognition of AIC in daily practice. More than a third of patients with newly diagnosed AF have heart failure (HF), and more than half of patients with newly diagnosed HF have concomitant AF [1]. AF and atrial flutter (AFlut) can cause LVSD and thus AIC, which is classified as a subset of dilated cardiomyopathy [2]. However, the causal relationship between tachyarrhythmia and LVSD is often difficult to assess, because any form of LVSD can lead to arrhythmia and vice versa—a classic “chicken-and-egg” situation [3]. As a hallmark of AIC, its phenotype can be treated with curative rhythm restoration [4,5,6]. Therefore, the diagnosis of AIC can only be made after successful therapy, which is known as diagnosis ex juvantibus. Arrhythmic heartbeats are the causal origin of AIC, and rhythm restoration is the causal therapy. It is important to note that AIC may occur in an already diseased heart, such as after a myocardial infarction. In this case, the impure form of AIC would be triggered by the arrhythmia. However, to characterize AIC and estimate the time to recovery from LVSD as accurately as possible, it is best to investigate the pure form of AIC that is due to the arrhythmia alone.

The recovery times reported in the literature are very variable and range from a few days to a several months [4,5,7,8,9,10]. This may be due to a heterogeneous patient population with differences in LV function, concomitant cardiac disease, method of rhythm control, and initial hemodynamic parameters, especially heart rate. Phenotypic changes in the LV itself such as structural remodeling with fibrotic changes could accompany AIC. Therefore, geometric and morphological criteria might be useful. In the study by Hsu et al. [4], data from cardiac magnetic resonance imaging (cMRI) and LV late gadolinium enhancement (LGE), as a marker of fibrosis, were not available. In the study by Muller-Edenborn et al. [5], contrast-enhanced cMRI could not predict the onset or likelihood of LVEF improvement after rhythm restoration. The CAMERA-MRI study evaluated the treatment effect of AF catheter ablation in patients with AF and idiopathic HF and found that the absence of LGE predicted greater improvements in absolute LVEF in patients undergoing catheter ablation [11]. In a registry, this patient population plus patients with paroxysmal AF as well as with ischemic heart disease underwent cMRI prior to pulmonary vein isolation [12]. Both studies showed better recovery of LVEF if no LGE was detectable.

The heterogeneity of patients studied thus far, with different types of AF coexisting with other types of heart diseases, may alter or obscure the arrhythmic effect and influence recovery from LVSD. The contribution of AIC in such a mixed population is difficult to evaluate. The objective of this study, as a substudy of our AIC trial [13], was to determine the recovery time in a homogeneous cohort of patients with pure AIC and evaluate whether recovery is dependent on the degree of LV fibrosis, measured as LGE on cMRI scans.

## 2. Materials and Methods

### 2.1. Study Population

In order to investigate the pure form of AIC, we screened for patients with tachyarrhythmia and LVSD in the emergency departments and inpatient and outpatient clinics of three hospitals in Germany (Figure 1A). Screening was performed by both electrophysiologists and cardiologists (general or interventional), depending on work schedule and site of presentation. Patients who were over 18 years old and who gave informed, written consent were included in the study if other causes of LVSD (such as valvular, ischemic, or inflammatory conditions) could be ruled out, LVSD was newly diagnosed, and AF or atrial flutter (AFlut) was persistent. Exclusion criteria for this study were paroxysmal or permanent AF/AFlut, relevant coronary stenosis identified by coronary angiography (either a diameter stenosis severity of ≥70% for non–left main disease and ≥50% for left main, left anterior descendent disease, and two- or three-vessel disease with LVEF <35%, or impaired coronary flow [fractional flow reserve ≤ 0.80, instantaneous wave-free ratio or resting full-cycle ratio ≤ 0.89], following current guidelines [14,15]), inability to participate in follow-up visits, recurrent arrhythmia during follow-up, development of other diseases that could potentially worsen LV function during follow-up, or a life expectancy of less than one year. Rhythm control was achieved following local clinical procedures with electrical cardioversion, ablation, and/or pharmacological intervention. After sinus rhythm had been established, LV geometry and function were assessed by LGE in cMRI. After six months of follow-up (visits every two months), patients were diagnosed with AIC if LVEF recovered significantly after 6 months (Figure 1B).

### 2.2. Echocardiographic Evaluation

Images were acquired using instruments from General Electric (GE), Solingen, Germany, or Philips Healthcare, Hamburg, Germany. Standard 2D parasternal and apical views were obtained. Grayscale image loops were recorded at frame rates of 50 to 70 frames/second. The sector width and depth were adjusted to capture the entire myocardium, including the epicardial surface. The image data were analyzed offline using IntelliSpace Cardiovascular Software release 5.1 (Philips Medical Systems, Hamburg, Germany). Blinding was unnecessary as the diagnosis was made after assessing the treatment success (diagnosis ex juvantibus). For analysis, we used the image loops with the highest quality and calculated LVEF using Simpson’s biplane method. During AF, we used the mean LVEF of five consecutive cardiac cycles.

### 2.3. Magnetic Resonance Imaging and Analysis

Following rhythm control, cMRI was performed using a Philips Achieva 3 Tesla (Philips Healthcare, Hamburg, Germany), a Siemens Avanto Fit 1.5 Tesla, or a Siemens Skyra 3 Tesla (Siemens Healthineers, Erlangen, Germany) clinical scanner. The sequence protocol was preceded by survey images in the coronal, sagittal, and transverse orientations. Retrospectively gated steady-state free precession cine images were acquired in the long and short axis orientations of the LV as described in Schach et al. [16] and analyzed as a substudy of our Clinical Characterization of AIC trial [13]. LGE was assessed 10 min after intravenous administration of 0.15 mmol/kg Gadobutrol (Gadovist^®^, Bayer, Leverkusen, Germany) using a short-axis stack and phase-sensitive inversion recovery sequence to identify regional fibrosis. IntelliSpace Cardiovascular Software release 5.1 (Philips Medical Systems, Hamburg, Germany) was used to analyze images from all study centers. Both qualitative and quantitative analyses were performed for LGE. LGE positivity was defined as LGE in at least one myocardial segment. Quantitative LGE analysis was performed in the short axis based on the endocardial and epicardial border contours from LV volume and mass measurement. Semi-automatic LGE measurement was corrected manually if necessary. Pathologic LGE was considered present if the LGE signal was more than three times greater than the standard deviation of a reference myocardial segment that was free of LGE. The results are expressed as a percentage of LGE in the LV myocardium.

### 2.4. Rhythm Control

Three strategies were used for rhythm control: (a) Electrical cardioversion (ECV) with optional pretreatment with antiarrhythmics to enhance the success of rhythmization, according to local clinical pathways was conducted as follows: transesophageal echocardiography was performed prior to ECV to rule out thrombus formation if oral anticoagulation was not yet started [17]. Patients were anesthetized with sevoflurane inhalation (8 vol%) until loss of consciousness. Blood pressure, heart rate, and oxygen saturation were monitored during the procedure. The Lifepak 15 (Stryker/Physio-Control Inc., Redmond, WA, USA) was used to deliver up to three synchronized shocks (300-360-360 J) [18] through self-adhesive electrodes in the anterior-posterior position until sinus rhythm was restored [19]. (b) Pulmonary vein isolation was performed as previously described [20]. Successful isolation of the pulmonary veins was confirmed by demonstrating input into and exit block out of the pulmonary veins. (c) Ablation of typical atrial flutter: The cavotricuspid isthmus was ablated by creating lesions from the most inferior part of the tricuspid valve to the ostium of the vena cava using a radiofrequency ablation catheter with an energy of 45 W. Bidirectional block was confirmed with differential pacing criteria.

### 2.5. Statistical Analysis

Data were examined for normal distribution using the Shapiro–Wilk test. Continuous variables are expressed as mean ± SD or as median (IQR), and categorical variables are provided as absolute numbers and percentages. Changes in echocardiographic parameters and biomarkers over time within the respective group (AIC or non-AIC) and comparisons between both groups were analyzed by two-way ANOVA (or mixed model) with Tukey’s (to analyze the influence of time course) and Šídák’s (influence of group) multiple comparison tests. Correlation analysis and simple linear regression were performed to evaluate the association between parameters and the strength of the model. For calculation of the time to recovery, a third order polynomial (cubic) curve fit was employed [21]. For evaluation of the performance of a cut-off value for LV LGE to predict time to recovery, simple regression analysis was performed. The cut-off point for optimal sensitivity/specificity was estimated by the Youden index [22]. Data were analyzed using standard statistical software (SPSS, version 26, IBM, Ehningen, Germany, and Graphpad Prism, version 9, San Diego, CA, USA) and a *p* value of <0.05 was considered statistically significant.

## 3. Results

### 3.1. Clinical Characteristics

Of the 68 patients initially enrolled, 29 were excluded for the reasons shown in Figure 1A. The study cohort consisted only of patients with the pure form of AIC, in which LVSD is caused solely by the arrhythmia itself. All patients were of European/white ancestry. Patients with any detected rhythm other than sinus rhythm were excluded. Finally, 39 patients (69% male, 67 ± 11 years) were analyzed, consisting of 31 patients with AIC (study group) and 8 patients without AIC (control group). Their baseline characteristics are provided in Table 1. There were no significant differences between the responders to rhythm control (AIC) and the non-responders (non-AIC).

### 3.2. Cardiac Magnetic Resonance Imaging

cMRI was performed after rhythm control at the initial presentation The results showed that non-AIC patients had larger end-diastolic and end-systolic volumes compared to AIC patients, despite having similar values of stroke volume and LVEF (Table 2). These findings suggest that non-AIC patients have advanced dilated LV remodeling.

### 3.3. Left Ventricular Geometry and Systolic Function over Time

Initial LV end-diastolic (LVEDD), but not end-systolic (LVESD) diameter, was smaller in AIC (*p* = 0.012 and *p* = 0.221 vs. non-AIC, respectively). At month 2, both LVEDD and LVESD differed between the groups. However, this difference was not detectable at months 4 and 6, when the diameters began to decrease slightly in non-AIC patients as well. By the end of follow-up, LVESD had decreased significantly after rhythm control in AIC, from 44.0 ± 7.1 mm to 37.7 ± 5.7 mm (*p* = 0.0004). However, LVEDD decreased at month 2 but did not show a significant change until the end of follow-up, from 54.3 ± 6.2 mm to 52.3 ± 6.4 mm (0.392), respectively (see Figure 2A,B). LVEF increased in both groups, with the strongest gain observed after 2 months, indicating an early differentiation at this time. The increase was 16.5 ± 7.9% in AIC and 7.0 ± 4.0% in non-AIC (*p* = 0.019). In AIC, LVEF continued to increase until the end of follow-up by 4.5 ± 1.1% (*p* = 0.002 vs. 2 months), whereas LVEF remained at the same level in non-AIC (change by 1.4 ± 1.3%, *p* = 0.896, Figure 2C).

Initial echocardiographic and cMRI data of LV geometry and systolic function showed a high correlation, with r values of >0.7 and *p* < 0.0001 (Figure 3). However, initial echocardiography was performed in patients with AF, whereas cMRI was conducted (for quality reasons) in sinus rhythm after the patients’ rhythm was restored. We observed lower values for LVEDD and LVESD measured via echocardiography, with a regression equation of Y = 0.615 × X + 18.5 for LVEDD and Y = 0.527 × X + 19.8 for LVESD. Correlation of LVEF had the highest values for r (0.814) and r^2^ (0.664), with Y = 0.751 × X + 6.3 as regression equation.

### 3.4. NT-proBNP and High-Sensitivity Cardiac Troponin T as Markers of Cardiac Stress

The levels of N-terminal pro b-type natriuretic peptide (NT-proBNP) declined over time in AIC (Figure 4A). In non-AIC, levels decreased less pronounced without reaching statistical significance; there was no effect between the groups. High-sensitivity cardiac troponin T (hs-cTnT) did not show any significant change over time or between the groups (Figure 4B).

### 3.5. LV Recovery Depends on LV LGE Content

The time to recovery (TTR) in AIC demonstrated a positive correlation with LV LGE as a measure of LV fibrosis in LGE-positive patients (see Figure 5A). In other words, shorter TTR is associated with less LV LGE. In these patients, mean LV LGE content was 8.3 ± 4.8% vs. 7.7 ± 3.8% in non-AIC patients (*p* = 0.779). As mentioned above, LVEF showed its greatest gain 2 months following rhythm restoration, with minor (AIC) or no changes (non-AIC) during the remaining follow-up. The gain in LVEF up to the second month (ΔEF), as a measure of rapid recovery, was also correlated with LV LGE (Figure 5B). There was no correlation between LV LGE and gain in LVEF up to month 6 (*p* = 0.224, r = −0.284, r^2^ = 0.081), which is consistent with full recovery from LVSD, independent of LV LGE. Regression equations were Y = 0.240 × X − 0.170 for TTR and Y = −0.997 × X + 27.2 for ΔEF. Grouping patients based on LV LGE < 13% vs. ≥13% showed the greatest difference in TTR from LVSD and had the most distinct separation at 1.7 months, highlighting the significance of this threshold for the timing of additional diagnostics. The performance of this binary group assignment is illustrated by the receiver operating characteristics curve, which has an area under the curve of 0.775 and a *p*-value of 0.016 (Figure 5C,D).

## 4. Discussion

In the current study collective, which was intensively evaluated by diverse diagnostics (e.g., every patient received coronary angiography, cMRI, and sequential echocardiography), we analyzed LV LGE content in a semi-automated manner and correlated it with (a) time to recovery from LVSD and (b) gain in LVEF in rhythm control responders, i.e., AIC patients. It is important to note that this collective is highly homogeneous, as our aim was to investigate the pure form of AIC. Therefore, we excluded all patients whose LVSD could be explained by other factors such as ischemia or inflammatory or valvular disease.

Previous trials investigating patients with AIC used varying upper limits of LVEF for patient inclusion, ranging from 40% to 50%. These limits correspond to the thresholds for HF with reduced and preserved LVEF [23,24]. However, consistent criteria for LVEF recovery have not been adopted across studies in this area, leading to potential inconsistencies in patient eligibility [11,16,25,26,27]. The definition used in our study is a composite of the criteria used in the studies cited above and was also utilized in the main trial (Clinical Characterization of AIC trial) [13]. There is significant confusion regarding the characteristics of the pure form of AIC in different clinical trials. Therefore, our study cannot be compared with large studies such as CASTLE-AF and EAST-AFNET4 due to the heterogeneous collectives in those trials that included patients with preexisting HF, coronary artery disease, and valvular disease [6,28].

To ensure accurate assessment of AIC patient recovery, it is crucial to distinguish their time course of recovery from patients without AIC, such as those with idiopathic cardiomyopathy or non-AIC. The CHAMP-HF cohort observed a 4% improvement in mean LVEF among patients with chronic HF after a median follow-up period of 16 months [29]. The PROVE-HF cohort, consisting of patients with new-onset HF (or without ACE inhibitors or angiotensin receptor blockers at baseline), showed a 7% improvement in LVEF within 6 months of initiating treatment with sacubitril/valsartan [30]. In other studies on myocardial recovery in patients with HF, the mean gain in LVEF in response to HF therapies ranged from 1.3% in 78 weeks (valsartan) to over 2.7% in 21 weeks (cardiac resynchronization therapy) and up to 12.0% in 52 weeks (bisoprolol) [31,32,33,34]. The follow-up of 6 months chosen in the current study was based on a previous ablation study on patients with HF that demonstrated an improvement of LVEF at month 6 following ablation without further gain thereafter [4]. In contrast to the aforementioned HF studies, patients diagnosed with AIC in our study showed an improvement in LVEF of 16.6 ± 8.1% after 2 months of rhythm restoration and 21.5 ± 7.1% after 6 months. This is at least a twofold improvement in LVEF and suggests that our diagnostic criteria for AIC were not too strict and thus focused on the correct patient population.

LVEF demonstrated the greatest improvement during the initial months of follow-up, consistent with the abovementioned ablation study by Hsu et al., and the cardioversion study by Mueller-Edemborn et al. which investigated LVSD recovery just 3 and 40 days after rhythm control but without additional follow-up [4,5]. In the latter study, cMRI with contrast did not predict recovery from LVSD, which contrasts with studies that have shown that the presence of LGE is associated with impaired recovery from LVSD [11,12]. Although there was no difference in LV LGE content between responders (AIC) to rhythm control and non-responders (non-AIC) in our study, we observed a time-dependent recovery that was influenced by LGE content, i.e., recovery took longer when LV LGE was higher. A lower LV LGE content was reported in patients recovering from LVSD in a registry of patients undergoing cMRI prior to pulmonary vein isolation [12], consistent with our findings. It is still unclear whether LGE, as a proxy for LV fibrosis, precedes LV dysfunction in arrhythmias or whether the arrhythmia itself leads to fibrosis. Until there is a longitudinal study of apparently healthy subjects who develop AF, this issue is difficult to resolve with certainty. In the case of atrial fibrosis, which was not measured in our study, there is evidence that fibrosis may precede AF in most cases (e.g., age-related fibrosis) [35,36], although exceptions may exist (e.g., lone AF) [37].

Cardiac MRI analysis in our study, which was performed in sinus rhythm, revealed a smaller LVEDD compared to the initial echocardiographic assessment in patients with AIC, which is similar to the echocardiographic observation at month 2. LV diameters seem to be more affected by rhythm restoration than by time alone. However, it takes several months in sinus rhythm until full recovery of LVSD is achieved. We also measured larger diameters and volumes in cMRI compared to the echocardiographic evaluation. This finding has also been demonstrated in other studies involving patients with AF or in sinus rhythm [38,39]. In addition, the high Y-axis intercept has been observed in previous correlation analyses of LVEF between cMRI and echocardiography [38,39,40]. The slope has only been reported in Guo et al. [38]. In this study, values for end-diastolic and end-systolic measures were similar to our data. Interestingly, echocardiography slightly underestimated the LV function compared with MRI in the latter study. Other studies have shown an increase in LVEF as an acute effect of electrical cardioversion [41,42]. Both findings explain the difference in LVEF we observed between cMRI and echocardiography.

### Limitations of the Study

Although prospectively conducted, the trial design is observational in nature due to the fact that AIC diagnosis only can be established after assessment of the treatment access of rhythm control (diagnosis ex juvantibus). The above criteria for the diagnosis of AIC were established in advance of the study Ref. [16]. Surprisingly, the use of these criteria resulted in an unexpectedly high prevalence of AIC in this cohort and a small control group of non-responders (non-AIC) consisting of only eight patients. Another limitation of the study is that baseline echocardiography was performed during AF instead of immediately after rhythm control due to the use of different methods and time points for administering the rhythm control strategy. Therefore, LV function was uniformly assessed at 2 months after rhythm control. While it is possible that LVEF may recover beyond the 6-month follow-up period, the overall time to recovery is expected to be within a few months, as discussed above.

## 5. Conclusions

In this well-characterized group of patients with newly diagnosed, unexplained LVSD and coexisting tachyarrhythmia, there was a positive correlation between the time required for LVEF recovery and the amount of LV LGE, used as a surrogate for fibrosis. These data may provide important diagnostic information to better tailor treatment strategies for these patients.

## Figures and Tables

**Figure 1 life-14-00330-f001:**
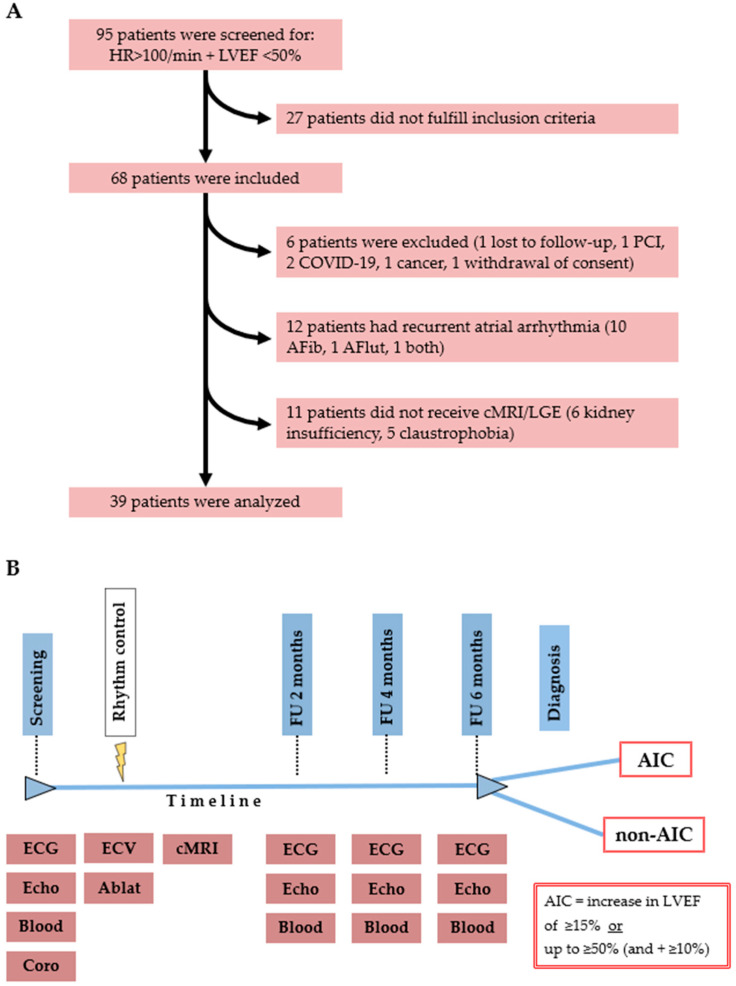
(**A**): Patient and study flow chart. (**B**): Study timeline. Patients were analyzed after screening and application of inclusion and exclusion criteria. Initially, electrocardiography (ECG), echocardiography (Echo), acquisition of blood samples (Blood), and coronary angiography (Coro) were performed. The dotted line indicates the position in the timeline and corresponds to the diagnostics mentioned in the boxes below the timeline. After rhythm control (represented by the yellow thunderbolt) by electrical cardioversion (ECV) and/or ablation (Ablat), cardiac magnetic resonance imaging (cMRI) was performed. Diagnostics were repeated every 2 months during follow-up until the end of follow-up at month 6 after rhythm control, and some patients were additionally monitored via Holter ECG. Thereafter, patients were classified as responders (AIC) if left ventricular ejection fraction (LVEF) improved significantly (either an increase of ≥15%, or an increase of ≥10% if above 50%) and distinguished from non-responders (all patients who did not fulfil these criteria, non-AIC). AFib = atrial fibrillation; AFlut = atrial flutter; AIC = arrhythmia-induced cardiomyopathy; COVID-19 = Coronavirus disease 2019; HR = heart rate; FU = follow-up; LGE = late gadolinium enhancement; PCI = percutaneous coronary intervention.

**Figure 2 life-14-00330-f002:**
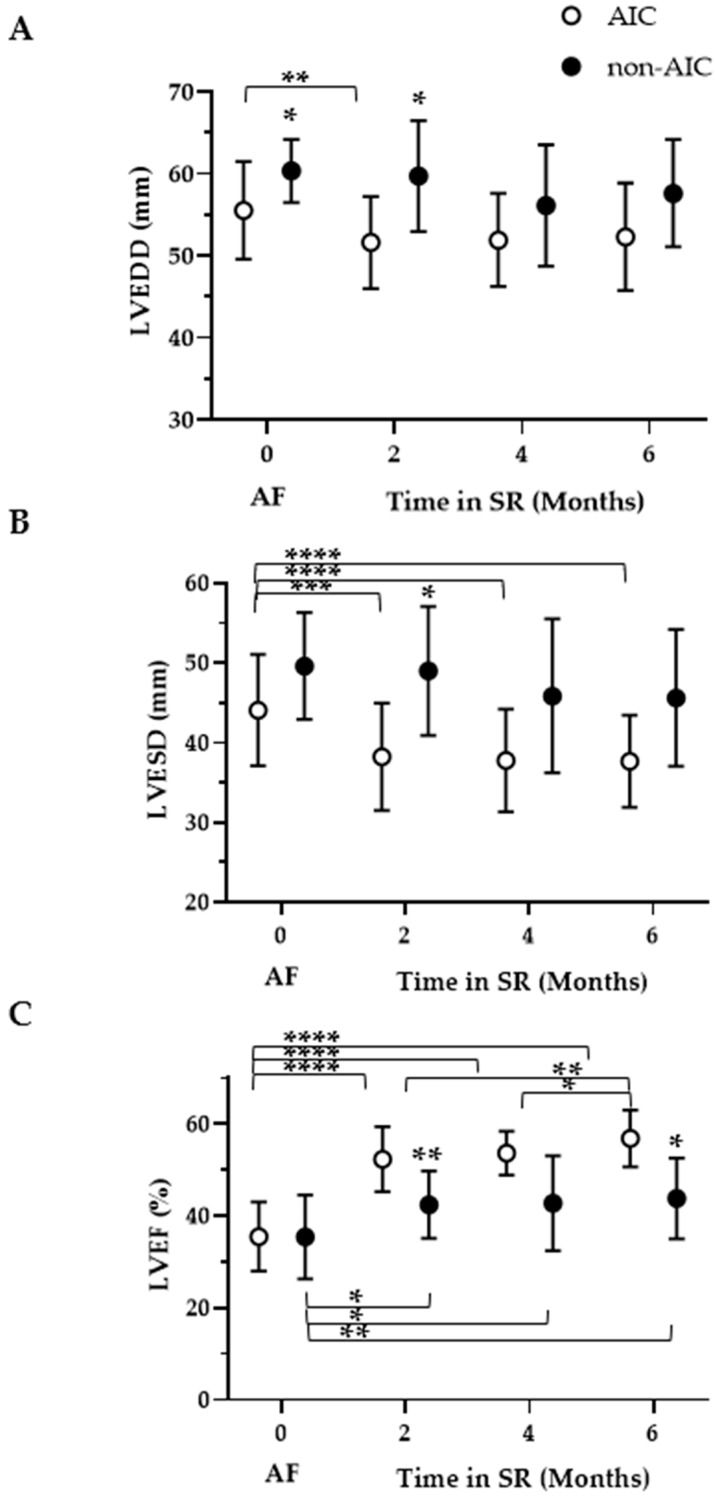
Left ventricular geometry and systolic function (echocardiography). Scatter plots of left ventricular end-diastolic diameter (LVEDD, (**A**)), left ventricular end-systolic (LVESD, (**B**)) and left ventricular ejection fraction (LVEF, (**C**)) plotted over time in patients with AIC (arrhythmia-induced cardiomyopathy, open circles) and non-AIC (solid circles); AF = atrial fibrillation/flutter; SR = sinus rhythm. * denotes *p* < 0.05 compared to the respective time period in the same group (brackets) or compared to the respective group when located above the upper error bar; the same applies to ** *p* < 0.01, *** *p* < 0.001, and **** *p* < 0.0001.

**Figure 3 life-14-00330-f003:**
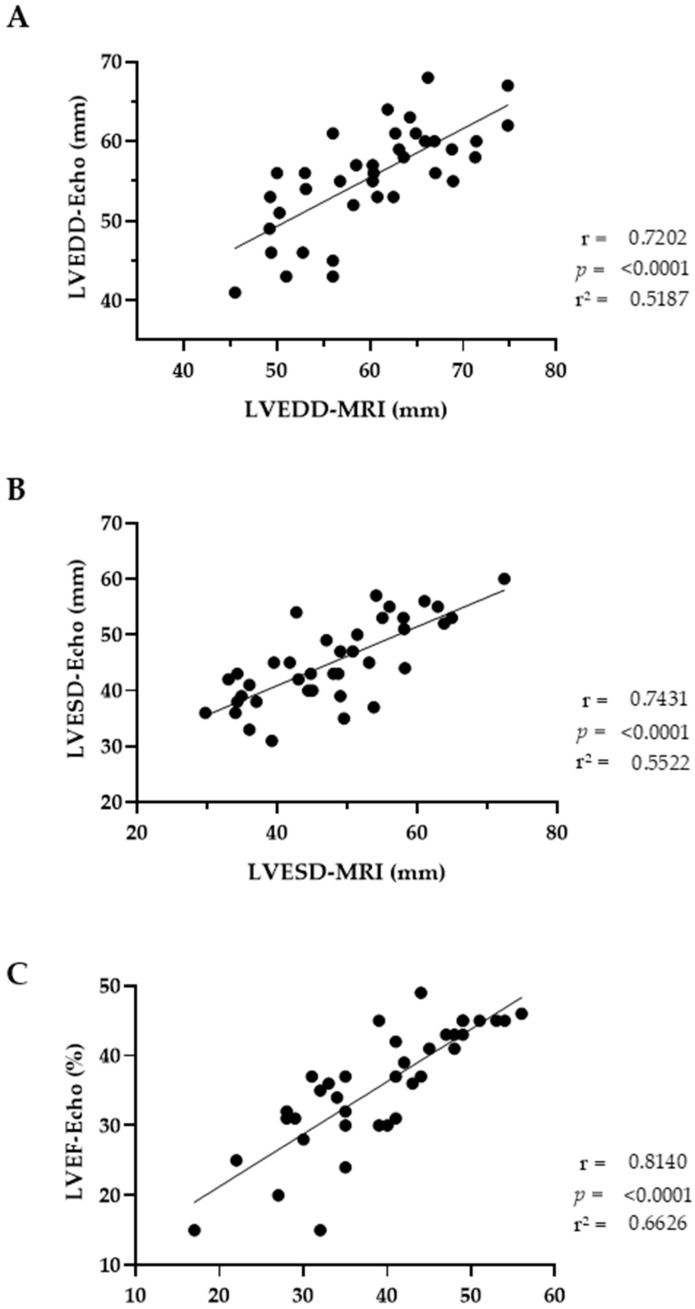
Correlation and simple linear regression of LV geometric and functional data between echocardiography and cMRI. Left ventricular end-diastolic diameter (LVEDD, (**A**)), end-systolic diameter (LVESD, (**B**)), and ejection fraction (LVEF, (**C**)) are compared between echocardiography (Echo) and magnetic resonance imaging (MRI). The black dots are individual values; the lines represent the best-fit values.

**Figure 4 life-14-00330-f004:**
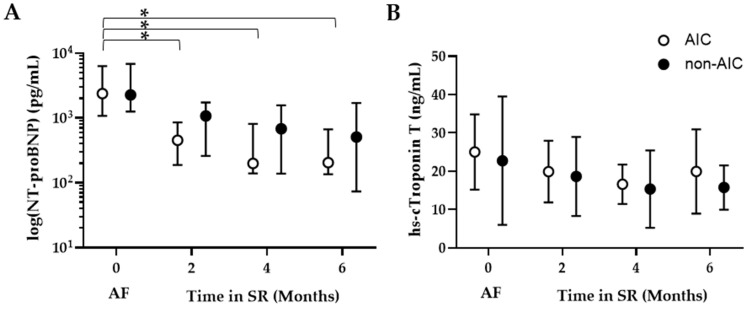
Scatter plots of biomarkers for cardiac stress in AIC (open circles) vs. non-AIC patients (solid circles). (**A**): N-terminal pro b-type natriuretic peptide (NT-proBNP, median with IQR, (**A**) and (**B**): High-sensitivity cardiac troponin T (hs-cTnT, mean with 95% CI). Abbreviations as in Figure 2. * denotes *p* < 0.001.

**Figure 5 life-14-00330-f005:**
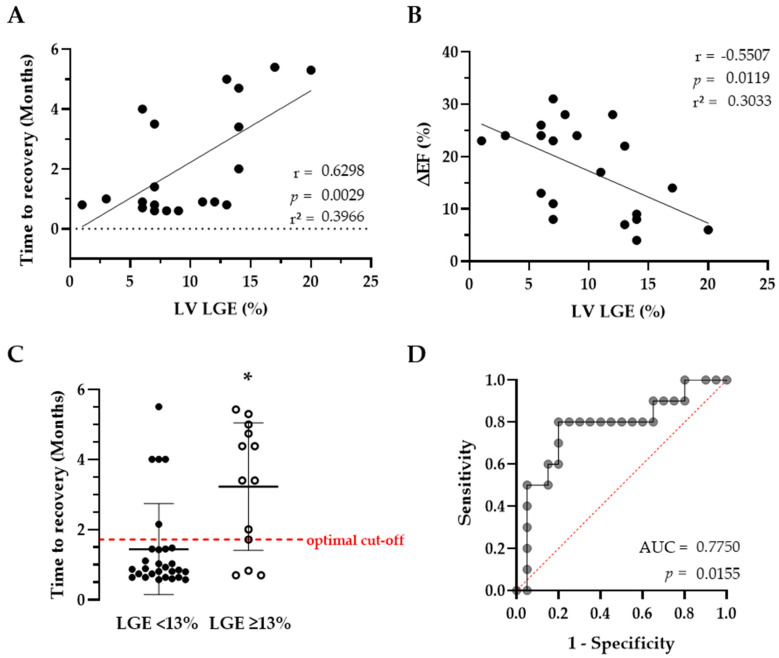
Correlation and simple linear regression modeling of the outcomes time to recovery (**A**) and ΔEF (month 2 minus baseline, (**B**)). (**C**): scatter plots with individual values, mean and SD. The red dotted line represents the point of time with the best separation between the groups. (**D**): receiver operating characteristics curve illustrating the performance of the group assignment (LGE < 13%) on the outcome time to recovery. * denotes *p* < 0.001. AUC = area under the curve; EF = left ventricular ejection fraction; LGE = left ventricular late gadolinium enhancement.

**Table 1 life-14-00330-t001:** Baseline characteristics of the study population.

Parameter	All (*n* = 39)	AIC (*n* = 31)	Non-AIC (*n* = 8)	*p* Value
Age	67.2 ± 11.2	67.6 ± 11.0	65.5 ± 12.3	0.510
Sex, male	27 (69)	21 (69)	6 (75)	0.692
Weight (kg)	92.1.0 ± 26.0	92.4 ± 22.8	90.9 ± 38.0	0.743
Height (cm)	174.6 ± 11.0	174.8 ± 11.1	173.8 ± 11.6	0.951
BMI (kg/m^2^)	29.7 ± 6.5	29.8 ± 5.9	29.4 ± 9.1	0.650
BSA (m^2^)	2.1 ± 0.3	2.1 ± 0.3	2.1 0.4	0.809
Diabetes	8 (21)	8 (26)	0	0.107
Arterial hypertension	29 (74)	24 (77)	5 (63)	0.389
Severe renal insufficiency *	11 (28)	9 (29)	2 (25)	0.821
CHA_2_DS_2_-VASc-Score	3.3 ± 1.7	3.4 ± 1.5	3.1 ± 2.4	0.299
Smoker status	5 (13)	4 (13)	1 (13)	0.976
Previous PCI/CABG	4 (10)	4 (13)	0	0.284
NYHA class	2.9 ± 0.6	2.9 ± 0.6	3.0 ± 0.7	0.544
LVEF (%)	35.4 ± 8.5	35.3 ± 8.3	35.6 ± 9.2	0.745
Heart rate (beats/min)	125.8 ± 16.5	127.1 ± 17.3	120.8 ± 11.9	0.362

Values are mean ± SD or n (%). AIC = arrhythmia-induced cardiomyopathy; BMI = body mass index; BSA = body surface area; CABG = coronary artery bypass graft; LVEF = left ventricular ejection fraction; NYHA = New York Heart Association; PCI = percutaneous coronary intervention. *p* values were calculated for AIC vs. non-AIC by unpaired t-test or Chi-squared test. * estimated glomerular filtration rate < 30 mL/min.

**Table 2 life-14-00330-t002:** LV geometric and functional cMRI data.

Parameter (n)	AIC (31)	Non-AIC (8)	*p* Value
LVEDD (mm)	58.7 ± 7.4	65.8 ± 4.2	0.009
LVEDV (mL)	165.1 ± 53.5	233.7 ± 72.4	0.002
LVEDV/BSA (mL/m^2^)	77.4 ± 21.4	115.4 ± 35.9	0.001
LVESD (mm)	45.9 ± 9.9	55.3 ± 7.5	0.012
LVESV (mL)	98.5 ± 46.4	156.4 ± 71.7	0.005
LVESV/BSA (mL/m^2^)	46.1 ± 20.0	77.5 ± 37.4	0.004
SV (mL)	66.7 ± 24.9	77.3 ± 22.3	0.264
SV index	31.3 ± 10.2	37.9 ± 8.2	0.181
IVSd (mm)	11.8 ± 2.2	11.3 ± 0.9	0.768
LVPWd (mm)	9.7 ± 1.8	9.5 ± 1.9	0.996
LVEF (%)	39.6 ± 9.0	38.0 ± 8.9	0.523

Data are presented as mean ± SD. Parameters are measured via blood analysis in cardiac magnetic resonance imaging (cMRI). AIC = arrhythmia-induced cardiomyopathy; BSA = body surface area; IVSd = interventricular septum thickness in diastole; LVEDD = left ventricular end-diastolic diameter; LVEDV = left ventricular end-diastolic volume; LVEF = left ventricular ejection fraction; LVESD = left ventricular end-systolic diameter; LVESV = left ventricular end-systolic volume; LVPWd = left ventricular posterior wall thickness in diastole; SV = stoke volume.

## Data Availability

The primary data analyzed in this study are not publicly available; some parameters that do not contain personal information can be provided by the corresponding author upon reasonable request.

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
