# Peer review of "Time to Recovery from Systolic Dysfunction Correlates with Left Ventricular Fibrosis in Arrhythmia-Induced Cardiomyopathy"

_life, 2024, doi:10.3390/life14030330_

Round 1

Reviewer 1 Report

Comments and Suggestions for Authors

1. The paper is nicely written .

2. Please include this paper in the reference (Severity of left ventricular dysfunction in patients with tachycardia0induced cardiomyopathy : Impact on remodeling after atrial flutter ablation . AJC ) .

3. The discussion could be shortened .

Reviewer 2 Report

Comments and Suggestions for Authors

This is a nice study of 39 pts with LV systolic dysfunction associated with tachyarrhythmia, divided according to response of the rhythm control therapy into AIC and non-responders.

Major issues (these should be mentioned in the limitations section):

- small groups (especially non-AIC n=8)

- baseline echocardiography would have been better to be considered immediately after rhythm control (similarly to cMRI), as long as fast HR during  examination might overestimate the degree of LV dysfunction

- follow-up up of only 6 months (although majority of AIC respond to rhythm control early, a significant percentage may respond till 1 year); considering that non-responders have significantly more dilated LV maybe a longer follow-up could reveal that some of them are still responding (especially because there was no difference in term of % of LV LGE between AIC and non-responders!)

Minor issues:

- introduction too long for a research article

Reviewer 3 Report

Comments and Suggestions for Authors

The authors should be congratulated for a novel and interesting study. My initial feedback is listed below for your point-by-point responses, which will eventually allow me to perform a complete review of the article's findings.

1. Do the authors believe that the degree of fibrosis actively/directly impedes the recovery process? Or perhaps the degree of fibrosis only correlates with the rate of recovery? The title, abstract, and body are written as though the fibrosis plays an active role in mediating recovery rate, despite the fact that fibrosis is the evidence of arrhythmia-induced cardiomyopathy, rather than the cause. Though, subtle, this difference in terminology/language is distinct, as words long "prolong" or "favor" suggest a mechanistic study, where as correlation suggests an association study.

2. It is ideal for the keywords to be distinct from the words of the title to increase the searchability and eventual readership of the article.

3. It is important to clearly describe who performed the screening of patients to be enrolled into this study, as selection bias is a concern. Were these individuals trained cardiologists, untrained medical assistants, both, neither?

4. Authors should explicitly state what p value was utilized to determine statistical significance.

5. There seems to be an error in the subtitle listed in line 174.

6. Figure 2, Figure 3, Figure 4, and Figure 5 contain an excessive proportion of empty white space. Optimization of the size of the panels would improve the quality of the figures. Where applicable, perhaps the authors can list the label legend once instead of beside each panel (excessively redundant) and place the panels next to each other.

7. It would be appropriate to report the race/ethnicity statistics for the cohorts, to improve the future external validity of these findings.

Round 2

Reviewer 3 Report

Comments and Suggestions for Authors

The reviewer thanks the authors for their excellent responses and revisions. A few minor critiques remains:

1. With the afore discussion regarding correlation versus causation, perhaps the title of the article should be revised regarding the use of "prolongs". This reviewer feels that it would be more appropriate to state something along the lines of "Extent of left ventricular fibrosis positively correlates with time to recovery in arrhythmia-induced cardiomyopathy". Also the type of correlation should be explicitly mentioned in the article (example: abstract conclusion, line 30).

2. In the abstract, it is not clear to me that the abbreviation cMRT is utilized after it is defined. Perhaps this unnecessary abbreviation can be eliminated.

3. I cannot find any publicly available data/publications regarding the "TACHY-AF Trial", if the authors desire to mention, a citation would be ideal. In addition, citations for for the CASTLE-AF and EAST-AFNET4 would be appreciated (and will likely increase the potential audience for the present article).

4. Not all abbreviations utilized in Tables 1 & 2 have definitions provided in the legends. Similarly additional abbreviations need to be defined within Figure 1 legend.
